# Asymmetric Conservation within Pairs of Co-Occurred Motifs Mediates Weak Direct Binding of Transcription Factors in ChIP-Seq Data

**DOI:** 10.3390/ijms21176023

**Published:** 2020-08-21

**Authors:** Victor Levitsky, Dmitry Oshchepkov, Elena Zemlyanskaya, Tatyana Merkulova

**Affiliations:** 1Department of System Biology, Institute of Cytology and Genetics, 630090 Novosibirsk, Russia; diman@bionet.nsc.ru (D.O.); ezemlyanskaya@bionet.nsc.ru (E.Z.); 2Department of Natural Science, Novosibirsk State University, 630090 Novosibirsk, Russia

**Keywords:** chromatin immunoprecipitation with massively parallel sequencing, transcription factors binding sites prediction, cooperative binding of transcription factors, composite elements, motifs conservation, classification of transcription factors, ETS transcription factor family, direct binding of transcription factors, overlap of motifs

## Abstract

(1) Background: Transcription factors (TFs) are main regulators of eukaryotic gene expression. The cooperative binding to genomic DNA of at least two TFs is the widespread mechanism of transcription regulation. Cooperating TFs can be revealed through the analysis of co-occurrence of their motifs. (2) Methods: We applied the motifs co-occurrence tool (MCOT) that predicted pairs of spaced or overlapped motifs (composite elements, CEs) for a single ChIP-seq dataset. We improved MCOT capability for the prediction of asymmetric CEs with one of the participating motifs possessing higher conservation than another does. (3) Results: Analysis of 119 ChIP-seq datasets for 45 human TFs revealed that almost for all families of TFs the co-occurrence with an overlap between motifs of target TFs and more conserved partner motifs was significantly higher than that for less conserved partner motifs. The asymmetry toward partner TFs was the most clear for partner motifs of TFs from the ETS (E26 Transformation Specific) family. (4) Conclusion: Co-occurrence with an overlap of less conserved motif of a target TF and more conserved motifs of partner TFs explained a substantial portion of ChIP-seq data lacking conserved motifs of target TFs. Among other TF families, conservative motifs of TFs from ETS family were the most prone to mediate interaction of target TFs with its weak motifs in ChIP-seq.

## 1. Introduction

Tissue-, cell- and stage-specific regulation of gene expression is produced through interactions of transcription factors (TFs) with respective regulatory elements called binding sites (BSs) or motifs; typically, each TF functions in tight cooperation with other TFs: there is a variety of mechanisms for cooperative TF–DNA binding [1,2]. Roughly, these mechanisms may be classified into simultaneous and sequential [1]. The first option implies a protein–protein interaction, and subsequent homo- or heterodimer binding to DNA. This mechanism may allow comparable or approximately equal impacts of affinity of two respective motifs. Alternatively, one TF of a pair may preliminarily interact with DNA, and at the second stage, ternary complex is formed through contributions of protein–protein and protein–DNA contacts of the second TF. This opportunity is facilitated by a higher DNA affinity of the first TF than for the second one. DNA-mediated interaction may also be facilitated by DNA conformation or nucleosomal organization [1], e.g., the propensity to interact with nucleosomal DNA is a special mark of pioneer TFs [3,4,5]. Thus, different mechanisms may explain a variety of possible TF–DNA ternary complexes, but in many cases, we may expect that behavior of two TFs is asymmetric. The recent review [6] proposed that in co-occurred pairs of motifs besides the orientation and spacing, the strength (affinity) of the individual motifs contributes to the specificity of a DNA regulatory region. Hence, systematic analysis of all possible partner motifs for various target motifs may propose the possible mechanism of cooperative TFs action.

Chromatin immunoprecipitation with massively parallel sequencing (ChIP-seq) analysis became the gold standard for protein/DNA-binding annotation at the whole genome level [7]. In particular, the ChIP-seq approach has been widely applied for the annotation of TFBSs; and the standard analysis pipeline at the final stage proposed application of de novo motif discovery tools that could confirm the presence of BSs specific for target (anchor) TF [7]. Since application of these tools for a single ChIP-seq datasets became a routine procedure [8], several attempts underlined the importance of massive analysis of motifs co-occurrence that reflected the cooperative mechanisms of TF actions [9,10]. We recently proposed the motifs co-occurrence tool (MCOT) package for motifs co-occurrence prediction in ChIP-seq data [11]. MCOT possesses two specific features, which are still absent in other analogous bioinformatics tools. First, MCOT uses a single ChIP-seq dataset for discovering motifs co-occurrence with a spacer and with an overlap. Second, MCOT performs simultaneous application of several thresholds for each motif; consequently, MCOT is able to retrieve composite elements (CEs) of anchor and partner motifs with various conservation ratios. Here the conservation of a motif implies its similarity to a recognition model.

In the current study we aimed to map anchor motifs in a benchmark ChIP-seq data for various TFs and predict which potential partner TFs might mediate their binding. We relied on estimation of the (a) co-occurrence of motifs for anchor and partner TFs and (b) asymmetry of motifs conservation in respective CE. In particular, we asked whether asymmetric pairs of anchor and partner motifs with more conserved partner motifs could explain earlier known substantial portions of ChIP-seq data lacking conserved anchor motifs (about a half of a ChIP-seq dataset, [12]). To investigate this issue we proposed the improvement for the MCOT computation procedure that directly reflected whether an observed misbalance between conservation of anchor and partner motifs was significantly higher than a random expectation. Consequently, for each pair of anchor and partner motifs, beside the conventional significance respecting CE enrichment, MCOT provided two additional significances that reflected enrichments of asymmetric CEs with more conserved anchor and partner motifs.

We carefully annotated anchor motifs for benchmark ChIP-seq data. Next, we calculated the abundance of CEs with a spacer and with an overlap for potential partner motifs from a library of known partner motifs. In particular, for each partner motif we separately analyzed asymmetric CEs with higher and lower conservation of partner motifs compared to respective anchor motifs. We classified all partner motifs according to families of partner TFs [13].

We concluded that only among overlapping pairs of anchor and partner motifs respecting all families of partner TFs, pairs with higher conservation of partner motifs were significantly more abundant than those with higher conservation of anchor motifs. Various TF families were differentiated according to the misbalance between asymmetric CEs with more conserved anchor and partner motifs. Thus, overrepresented asymmetric CEs with more conserved partner motifs and less conserved anchor motifs systematically promoted weak direct interactions of anchor TFs in ChIP-seq data. Among other families, partner motifs of TFs from the ETS family had the greatest misbalance in conservation toward partner motifs. Hence, we have shown that motifs of TFs from the ETS family systematically mediate cooperative binding of other TFs through higher conservation of ETS-like motifs in widespread CEs with an overlap of motifs.

## 2. Results

### 2.1. Integration of CE Significance and CE Asymmetry in the MCOT Analysis

We earlier developed the MCOT package for the prediction of spaced and overlapped pairs of co-occurred motifs in a single ChIP-seq dataset [11]. To perform the search of CEs, MCOT required the ChIP-seq dataset (peaks), the anchor motif that refers to the target TF and either the partner motif or the assignment of a public library of proven partner motifs; in the current study, we classified partner motifs from the Hocomoco library [14] according to the respective families of partner TFs (Figure 1A).

We applied a model of the position weight matrix (PWM) for motifs recognition. Besides the classification of CEs by the orientation, we classified them into fully/partially overlapped and spaced. We considered all orientations together and we updated the CEs classification according to motifs conservation (Figure 1B). The analysis of a scatterplot between conservation of anchor and partner motifs may reveal an extent of misbalance between similarities to recognition models of their motifs, more specifically the value −Log_10_(FPR) is the measure of motif’s conservation, here FPR denotes the false positive rate (Section 4.2).

Basic MCOT output data represents the significance of CEs regardless conservation of motifs in a pair and those for CEs with more conserved anchor or partner motifs (Figure 1C, Table 1). Thus, separate analysis of CEs with an overlap of motifs and with a spacer, the detailed classification of CE types, integration of homologous partner motifs of the same family and massive analysis of benchmark ChIP-seq data respecting various anchor motifs allowed to appreciate abundances of structurally specific CEs with partner motifs of various families (Figure 1D).

Additionally, in this study for each pair of motifs we proposed the significance of CE asymmetry (Table 2), which for a pair of motifs compared the content of asymmetric CEs with more conserved one motif with that for more conserved another motif. Table 1 and Table 2 show 2 × 2 contingency tables that illustrate the application of Fisher’s exact tests.

### 2.2. Single ChIP-Seq Dataset: Example of Significant Asymmetry within CE

In this section we illustrated the calculation of the asymmetry (Table 2 and Section 4.3) for CEs of the anchor FoxA2 motif (ChIP-seq dataset from mouse liver tissue [15]) and potential partner motifs from the Hocomoco mouse core collection, [14]. In the original study [15], besides the enrichment of anchor FoxA2 motifs, the authors revealed its co-occurrence with potential BSs of partner TFs GATA4, PAX6 and HNF1. The MCOT analysis confirmed significant co-occurrences of all respective CEs. However, only for HNF1β (HNF1B_MOUSE.H11MO.0.A, [14]) we found the extremely significant asymmetry within predicted CEs toward the partner motif (*p* < 2 × 10^−28^ and *p* < 4 × 10^−17^ for CEs with an overlap of motifs and with a spacer, respectively). Figure 2 shows the difference between relative frequencies of observed and expected CEs with specific conservation of FoxA2 and HNF1β motifs for their overlapped and spaced positioning.

MCOT analysis of other ChIP-seq datasets for FoxA2 and it close homologue FoxA1 revealed that FoxA1/2-HNF1β CEs with an overlap of motifs were significant, in some cases a moderate significance was also found for respective CEs with a spacer. However, the significant asymmetry in these CEs was not observed for other FoxA1/2 ChIP-seq datasets (FoxA2 for liver cancer cell line HepG2 [16]; GSM686926, FoxA1, prostate cell line LNCaP [17] and GSM1505633, FoxA1, embryonic cell lines, [18]). Thus, CE asymmetry toward the HNF1β motif appeared to be the specific feature of FoxA2-HNF1β CEs in liver tissue.

### 2.3. Single ChIP-Seq Dataset: Multiple Partner TFs Support Binding of Anchor TF

The application of MCOT may provide a list of potential partner motifs with the designation of relationships between conservation of motifs in a pair. The previous section represented a sole example of CE that had a higher conservation of a partner motif than an anchor motif. In practice, multiple partner TFs may cooperate with an anchor TF, and this may be in respect to several asymmetric CEs with more conserved partner motifs (and less conserved anchor motif). This may be a possible explanation of the absence of known motifs of anchor TFs in about a half of the peaks [12,19].

We previously showed that at least for FoxA2 in two ChIP-seq datasets [15,16] almost 100% of peaks contained potential motifs of anchor TF, although this conclusion was deduced due to an alternative to the PWM recognition model [20]. Consequently, the majority of FoxA2 peaks should contain at least moderately conserved FoxA2 motifs. Hence, we considered the same FoxA2 dataset [15] and excluded from analysis 37.7% of all (4455) peaks that had the most conservative hits (FPR respecting best hits in peaks below 5.24 × 10^−5^) and 19.1% of peaks that had too weak conservation of FoxA2 best hits (FPR above 5 × 10^−4^) (Section 4.2). The rest, 43.2%, of the peaks had FoxA2 hits with a moderate or weak conservation, 5.24 × 10^−5^ < FPR < 5 × 10^−4^. We believed that a portion of these peaks should contain CEs with respect to various more conserved partner motifs. To check that possibility, we performed a MCOT analysis and required that each partner motif beside the absence of similarity to the FoxA2 motif should have the significance of asymmetric CEs toward a partner motif. Thus, we selected the top 30 motifs according to the respective CE significance and sorted them by the fraction of peaks containing asymmetric CEs with more conserved partner motifs (Table 1, Section 4.2 and Section 4.3). Figure 3A represents the ranging of these partner motifs. As we expected, almost all peaks of the fraction with moderately and weakly conserved FoxA2 hits (89.6%) contained significant CEs with more conserved partner motifs. The first ranked motif FoxQ1 belonged to the same Forkhead box (FOX) factors {3.3.1} family as the FoxA2 motif, these motifs were moderately similar (*p* < 0.1), i.e., the MCOT filter detected their homology as not significant. Among other top-ranked partner motifs we found the BSs for previously known co-factors HNF1α/β and HNF6 (Figure 3A). The similarity filter excluded from our analysis motif HNF4γ (HNF4G_MOUSE.H11MO.0.C). Further analysis of motifs similarity within top 30 partner motifs (Figure 3B) demonstrated that they respected to relatively small numbers of TF families [13]. Thus, besides the first ranked FoxQ1 that belonged to the Forkhead box (FOX) factors {3.3.1} family, the next eight top-ranked motifs belonged to five families:
Thyroid hormone receptor-related factors (NR1){2.1.2} (NR1H3_MOUSE.H11MO.0.A),POU domain factors {3.1.10} (HNF1A_MOUSE.H11MO.0.A and HNF1B_MOUSE.H11MO.0.A),HD-CUT factors {3.1.9} (HNF6_MOUSE.H11MO.0.A and CUX2_MOUSE.H11MO.0.C),C/EBP-related {1.1.8} (NFIL3_MOUSE.H11MO.0.C),SOX-related factors {4.1.1} (SOX9_MOUSE.H11MO.0.A and SOX10_MOUSE.H11MO.0.B).


Notably, asymmetric CEs FoxA2/HNF1β, FoxA2/HNF6 and FoxA2/Sox9 with respect to different structural types of FoxA2, which could be represented by TATTTATTTA, TATTGACT and TGTTT(A/G)(C/T) (Appendix A), i.e., each time the FoxA2 motif is ‘adopted’ by a partner motif.

In total, ten top-ranked asymmetric CEs with more conserved partner motifs were contained in 29.5% of all ChIP-seq peaks and in 68.2% of peaks with moderately or weakly conserved FoxA2 motifs with FPRs from 5.24 × 10^−5^ to 5 × 10^−4^ (Figure 3). Accounting for 30 top-ranked asymmetric CEs increased these fractions up to 37.3% and 86.4%, respectively. Thus, a substantial portion of the FoxA2 ChIP-seq dataset [15] contained asymmetric CEs with more conserved partner motifs.

### 2.4. Massive Analysis of Asymmetric CEs

#### 2.4.1. Analysis of Partner Motifs Classified According to the TFs Families

In the previous section we performed an analysis of a single ChIP-seq dataset and approved that multiple motifs of various known and presumed partner TFs might be located near weak motifs of anchor TF (Figure 3). We asked whether certain partner motifs tended to mediate binding of various anchor TFs through asymmetric CEs with more conserved motifs of partner TFs. We took in an analysis of the benchmark data of 119 ChIP-seq datasets for 45 human TFs with annotated occurrences of anchor motifs and applied the library of 396 partner motifs from the Hocomoco database (Section 4.1). We applied the MCOT package for the prediction of CEs regardless of motifs conservation, and for CEs with more conservative anchor or partner motifs (Section 4.2 and Section 4.3, Table 1). Abundances of these types of CEs for full, partial, overlap, spacer and any computation flows for all partner motifs are given in Appendix A.

Since MCOT operated motifs, but not TFs, and the Hocomoco collections contained hundreds of more or less homologous motifs of various TFs, we organized all accepted in analysis 396 partner motifs into 50 clades according to recent classification of human TFs by the structure of DNA-binding domains [13]. These clades comprised of 49 families of TFs, and also one additional subfamily of CTCF-like motifs according to previous results [12] (Section 4.5).

The integrated MCOT application for all 119 anchor and 396 partner motifs produced 3623/4228 and 4484/14718 asymmetric CEs with more conserved anchor/partner motifs for “Full” and “Overlap” computation flows, respectively; the rest of the flows revealed substantially lower amounts (45/519, 287/56 and 499/15 for “Partial”, “Spacer” and “Any” flows, respectively; Appendix A). The Welch’s *t*-test for the number of ChIP-seq datasets possessing asymmetric CEs toward partner motifs vs. that possessing asymmetric CEs toward anchor motifs demonstrated the significance for “Full”, “Partial” and “Overlap” computation flows (*p* < 0.02, *p* < 1 × 10^−34^ and *p* < 1 × 10^−170^, respectively; Appendix A). The computation flows “Spacer” and “Any” revealed the significance in the reverse direction, i.e., asymmetric CEs toward anchor motifs were more abundant than those toward the partner motifs (*p* < 1 × 10^−9^ and *p* < 1 × 10^−41^, respectively; Appendix A). Thus, the higher conservation of partner motifs in CEs with an overlap of motifs has a systematic behavior, and abundance of such asymmetric CEs is substantially higher than that for CEs with a spacer. Hence, the focus in the consequent analysis will be on CEs with an overlap of motifs.

Figure 4 compares the number of ChIP-seq datasets containing asymmetric CEs toward partner motifs and that for asymmetric CEs toward anchor motifs for 50 selected above clades of TFs for the benchmark ChIP-seq data.

Since points for all clades in Figure 4 lie above the diagonal from the lower left to upper right (dashed line), we concluded that for all clades of partner motifs the abundance of CEs with asymmetry toward partner motifs exceeded that of CEs with asymmetry toward anchor motifs. The clades of partner TFs that were the most specific for asymmetry toward partner motifs with respect to families of ETS-related factors {3.5.2} and heteromeric CCAAT-binding factors {4.2.1} (points close to the top left corner, Figure 4). The families of p53-related factors {6.3.1}, RFX-related factors {3.3.3} and thyroid hormone receptor-related factors (NR1) {2.1.2} showed high abundance of both asymmetric CEs toward the anchor and partner motifs, since their points were close to the diagonal in Figure 4. THAP11 and CTCF-like motifs had a tendency to form asymmetric CEs toward the anchor motifs, since the top six clades for asymmetric CEs toward the anchor motifs were p53-related factors {6.3.1}, THAP-related factors {2.9.1}, CTCF-like factors {2.3.3.50}, thyroid hormone receptor-related factors (NR1) {2.1.2}, nuclear factor 1 {7.1.2} and RFX-related factors {3.3.3} (Figure 4).

To estimate for the benchmark data the enrichment of asymmetric CEs toward partner motifs vs. those toward anchor motifs we applied the Welch’s *t*-test for the counts of respective ChIP-seq datasets. Figure 5 shows the significance of this test as a function of the number of datasets containing CEs with an overlap of motifs (and regardless motifs conservation). Application of Bonferroni’s correction to set a threshold for the significance, *p* < 0.05/50 = 0.001 (Figure 5, dashed line) resulted in 45 out of 50 clades (90%) possessing the significant enrichment of the number of ChIP-seq datasets with asymmetric CEs toward the partner motifs. We found that the ETS-related factors {3.5.2} family combined

High abundance of CEs (axis X in Figure 5),Significant enrichment of asymmetric CEs toward partner motifs vs. those asymmetric toward anchor motifs (axis Y in Figure 5);High abundance of asymmetric CEs toward partner motifs in comparison with that for asymmetric CEs toward anchor motifs (Figure 4).

The majority of clades (26 out of 50) possessed the high significance, *p* < 1 × 10^−10^ (Figure 5, axis Y). The top four clades were FTZ-F1-related receptors (NR5) {2.1.5}, heteromeric CCAAT-binding factors {4.2.1}, ETS-related factors {3.5.2} and NGFI-B-related receptors (NR4) {2.1.4} (*p* < 1 × 10^−273^, *p* < 1 × 10^−198^, *p* < 1 × 10^−88^ and *p* < 1 × 10^−66^, respectively). The top twelve clades included also C/EBP-related factors {1.1.8}, CTCF-like factors {2.3.3.50}, Maf-related factors {1.1.3}, Jun-related factors {1.1.1} and Forkhead box (FOX) factors {3.3.1} (Figure 5). The differences were not significant only for five TF families: B-ATF-related factors {1.1.4}, factors with multiple dispersed zinc fingers {2.3.4}, GATA-type zinc fingers {2.2.1}, HD-CUT factors {3.1.9} and THAP-related factors {2.9.1}.

#### 2.4.2. Analysis of Top-Ranked Partner Motifs Classified According to TFs Families

In this section, we performed the detailed analysis of concrete top-ranked partner motifs participating in asymmetric CEs with more conserved partner motifs. This analysis was motivated by occasionally observed imperfect homology of motifs within separate families of TFs (Section 4.5), i.e., analysis of the previous subsection aimed be verified by top-ranked predictions for concrete motifs from various top-ranked TF families (Figure 4 and Figure 5). In addition, we should verify the MCOT results with the previous analysis [12] that revealed Jun-like, ETS-like, CTCF-like and THAP11 overrepresented motifs for the fraction of ChIP-seq data lacking canonical motifs of anchor TFs.

Thus, initially we checked the abundance of partner motifs participating in CEs regardless of the conservation of two motifs. We applied MCOT and selected 30 top-ranked partner motifs from the Hocomoco human core collection [14] motifs, excluding homologous pairs anchor–partner (Section 4.5) and performed the motifs clustering (Figure 6). Besides the Jun-like and ETS-like motifs, in the list of top-ranked partner motifs we found RFX-like motif, two motifs from Thyroid hormone receptor-related factors (NR1) {2.1.2} family, three GATA-like and three p53-like motifs (for description of these motifs Section 4.5). In Figure 6, we marked several families of TFs that were mentioned earlier by Worsley Hunt and Wasserman [12], or revealed above in our analysis (Figure 4 and Figure 5).

Among the 30 top-ranked motifs (Figure 6) we found many BSs of TFs from the two largest families (more than 3 adjacent zinc finger factors {2.3.3} and factors with multiple dispersed zinc fingers {2.3.4}, with 76 and 20 motifs, respectively; Appendix A). These families belonged to the C2H2 zinc finger TF class [13] with the highest known diversity of DNA binding specificities [21] and the lowest specificity in the benchmarking comparison with motifs with respect to other families [22]. Notably, the third largest family ETS-related factors {3.5.2} respected to 19 motifs, for these motifs the high homology was detected (Section 4.5, [23]) and good performance in benchmarking comparisons with motifs with respect to other families [22].

In general, the results of our analysis (Figure 6) are in good accordance with the previous analysis of Worsley Hunt and Wasserman [12]. Thus, ETS-like and Jun-like motifs were found among the top 30; however, we did not detect CTCF-like and THAP11 motifs, but still we previously found them among top-ranked TF clades (Figure 4 and Figure 5).

Next, we selected the 30 top-ranked partner motifs that formed asymmetric CEs toward either anchor or partner motifs, again we excluded homologous anchor–partner pairs and performed the motifs clustering (Figure 7). Notably, the separate analysis of asymmetric CEs toward the partner motifs had shown the larger variety than that for asymmetric CEs toward the anchor motifs (compare colored frames on panels A and B of Figure 7). Thus, NR1H3-like, RFX-like, GATA-like and p53-like motifs were found in both lists. Jun-like motifs, which we expected from the previous study [12], were absent in both lists. The rank of the best Jun-like motif MAF_HUMAN.H11MO.0.A was only 62 (Appendix A, the column “Conservative partner, Overlap”).

As for Jun-like motifs, our analysis that took into account the conservation of motifs (Figure 7) seemed to be contradictory to the one regardless of motif conservation (Figure 6). However, the previous study [12], which revealed overrepresented Jun-like motifs for the fraction of ChIP-seq data lacking canonical anchor motifs, did not check the homology between anchor and partner motifs. Hence, we canceled the restriction on the significant homology between CE participants and confirmed that the rank of Jun-like motifs substantially increased (Appendix A). Hence, we presumed that this enrichment of partner Jun-like motifs at least partially was based on their significant similarity to anchor motifs. Thus, we could not confirm the critical importance of Jun-like TFs in cooperative binding with other TFs to DNA.

Motifs of the ETS-related factors {3.5.2} family were found only in the list of asymmetric CEs toward partner motifs (Figure 7). Consequently, ETS-like motifs had the clearest tendency among motifs with respect to other families to form asymmetric CEs with less conserved anchor motifs so that within these CEs the similarity between anchor and partner motifs was absent.

All results presented above corresponded to the MCOT computation flow “Overlap”. The respective analysis of asymmetric CEs toward partner motifs for other computation flows revealed lower abundances of asymmetric CEs (Appendix A). In particular, the “Full” computation flow was shown with only two NR1H3-like motifs from the thyroid hormone receptor-related factors (NR1) {2.1.2} family (45 and 30 datasets, Appendix A) and three p53-like motifs (33, 30 and 29 datasets; Appendix A). In the “Partial” computation flow we revealed the first-ranked CTCF-like motif from the CTCF-like factors {2.3.3.50} subfamily, it was detected in only 8 ChIP-seq datasets, while for the spacer computation flow the first three motifs were NFYA-like (heteromeric CCAAT-binding factors {4.2.1} family) with respect to only 7, 6 and 6 datasets (Appendix A).

p53-like, GATA-like and NR1H3-like motifs have shown the enrichment in both cases of asymmetry toward the anchor and partner motifs (Figure 7). Thus, among other families, motifs of the ETS family most clearly demonstrate a specific enrichment in asymmetric CEs toward partner motifs. Hence, we may suppose that ETS-like motifs facilitate weak direct interaction of anchor TFs with their cognate binding sites in ChIP-seq peaks. In this case, the ternary complex {anchor TF, TF from ETS family, DNA} is formed so that the ETS-like motif is systematically more conserved than anchor motifs, i.e., TFs from the ETS family have the leading role in the cooperative interaction with other TFs, when they bind to DNA.

## 3. Discussion

Many studies confirmed that ChIP-seq data possessed a substantial portion of binding regions lacking the conserved motifs of target TFs [12,19]. In the current study, we aimed to clarify whether this portion might respect weak binding motifs of anchor TFs that were located near relatively more conserved motifs of multiple partner TFs. We applied recently a developed MCOT package for prediction of motifs co-occurrence with their overlaps and with spacers in a single ChIP-seq dataset [11]. The novelty of our study consisted of analysis of specific CEs with higher conservation of either anchor or partner motifs. We improved the previous algorithm [11] for estimation of the significance of such asymmetric CEs (Figure 1C, Table 1) and developed the novel methodology to measure the asymmetry within CEs toward one of the participant motifs (Table 2). Next, we have shown the example of the significant asymmetry within earlier known CEs FoxA2-HNF1β for ChIP-seq dataset from the liver tissue (Figure 2, [15]). The higher conservation of the HNF1β motif in these CEs proposed its leading importance in cooperative binding of both TFs, e.g., presumably HNF1β binding sites were preliminary occupied by HNF1β. This hypothesis is supported by the earlier observation that TFs HNF1β and FoxA3 are sufficient to reprogram mouse embryonic fibroblasts into induced hepatic stem cells [24]. The next example (Figure 3) illustrated the action of multiple partner motifs co-occurring near anchor FoxA2 motifs in the same ChIP-seq dataset [15]. We specifically excluded peaks with the most conservative and too weak FoxA2 motifs from the analysis. Peaks with the most conservative anchor motifs probably with respect to direct FoxA2 targets, too weak FoxA2 targets potentially required an alternative to the PWM model [20], so that we expected an expressive support for FoxA2 binding from partner TFs for intermediate cases of moderately or weakly conserved FoxA2 motifs. Our analysis demonstrated that about 90% of the analyzed peaks contained asymmetric pairs of co-occurred anchor and partner motifs, so that partner motifs possessed the higher conservation in pairs (Figure 3). Conventionally, in almost all analyses before MCOT, a single threshold for a recognition model of anchor motif was applied, so that weak interactions might be missed by a standard recognition model. Hence, these weakly conserved anchor motifs probably were annotated as indirect or non-specific binding (e.g., in [12,18]). We proposed that in this case multiple overrepresented asymmetric CEs with higher/lower conservation of partner/anchor motifs explained the absence of the most conserved motifs of anchor TFs in a substantial portion of peaks.

Next, we performed a massive analysis with the benchmark ChIP-seq data to study whether partner TFs from specific families possessing common characteristics of DNA-binding domains [13] tended to form specific asymmetric CEs toward partner motifs. As follows from the previous example (Figure 3), such partner TFs might have specific opportunities to mediate systematically the interaction of anchor TFs with their cognate binding sites in ChIP-seq data. Previously, Worsley Hunt and Wasserman [12] for the benchmark ChIP-seq data demonstrated that CTCF-like, Jun-like, ETS-like and THAP11 motifs had overrepresented motifs near summits in peaks lacking the canonical motifs of anchor TFs. These enriched motifs were termed “zingers” to highlight their outstanding enrichment in ChIP-seq datasets for various anchor TFs. With this knowledge, we took our benchmark data of 119 ChIP-seq datasets for 45 distinct TFs (Appendix A, [10]) with manually annotated anchor motifs derived from the de novo motif search [8] and predicted CEs with several additional criteria. In particular, we searched CEs that (a) respected a higher conservation of partner motifs than that of anchor motifs, and (b) did not respect the significant similarity between anchor and partner motifs. We proposed that the enrichment of such asymmetric CEs with simultaneously less significant enrichment of the respective CEs with higher conservation of anchor motifs, reflected a leading role of partner motifs in cooperative interaction of anchor/partner TF pairs with genomic DNA. Thus, we used a similar research strategy as Worsley Hunt and Wasserman [12], but our MCOT algorithm with varied thresholds of both motifs until the very loose (FPR = 5 × 10^−4^) allowed us to deduce potential CEs that were almost imperceptible with canonical threshold occurrences of anchor motifs. Moreover, our tool had the advantage for analysis of the co-occurrence of motifs with an overlap, which have been missed in previous studies for a single ChIP-seq dataset [9,10,25,26]. Additionally, the conventionally applied masking procedure (e.g., in [12]) for anchor motifs inevitably destroyed overlapping partner motifs, though overlapping of motifs were observed notably higher than their co-occurrence with a spacer [10,11,27].

Our results substantially extended and supplemented the previous study [12] (Figure 4, Figure 5, Figure 6 and Figure 7); we confirmed earlier conclusions concerning CTCF-like, Jun-like, ETS-like and THAP11 motifs. However, our analysis brought many details concerning specific families of TFs. Thus, we explained the enrichment of Jun-like motifs by their similarity to anchor motifs (Appendix A and Figure 7). Additionally, we found partner motifs of TFs from THAP-related factors {2.9.1} among the top-ranked in the list with respect to CEs with arbitrary conservation of motifs (Figure 6) and in the list with respect to asymmetric CEs toward anchor motifs (Figure 7A). Moreover, the THAP-related factors {2.9.1} family was detected among only five families among a total of 50 clades that did not possess the significant enrichment of the abundance of asymmetric CEs toward partner motifs vs. that for asymmetric CEs toward anchor motifs (Figure 5).

The detailed analysis (Figure 4, Figure 5 and Figure 7) demonstrated that besides the proposed earlier [12] CTCF-like, Jun-like, ETS-like and THAP11 motifs, other motifs, in particular NR1H3-like, RFX-like, p53-like, NFYA-like and GATA-like also systematically promoted binding of anchor TFs in ChIP-seq data. We may conclude that ETS-like motifs comprised of CEs with their highest conservation relative to anchor motifs, with respect to CEs were not enriched in the list of top ranked predictions for asymmetry toward anchor motifs, and ETS-like motifs were not significantly similar to anchor motifs participating in significantly enriched CEs.

The family of ETS-related TFs in human consists of 28 members [21], which are further classified into several subfamilies [13,21,23]. According to the comparative analysis of human TFs [21], besides the ETS family, only several other TF families or superfamilies, e.g., nuclear receptors, STAT and T-box, had the complete coverage of known motifs and absence of secondary motifs.

Recent all-against-all benchmarking of PWM models [22] suggested that the majority of ETS members have indistinguishable DNA binding specificity according to in vitro HT-SELEX assays. Thus, while a single PWM for ELK1 (MA0028.2 from JASPAR) was the best predictor for multiple TFs from the ETS family for in vivo and in vitro experiments; this matrix also was the best performer for ChIP-seq in vivo experiments for ten TFs, only five of which were ETS family members [22]. For the rest five unrelated TFs authors proposed the recruitment to their target binding sites through protein–protein interactions with a DNA-bound ETS factor. This hypothesis is in excellent accordance with our results (Figure 4 and Figure 5).

The previous analysis of genome binding of ETS family members [23] proposed that DNA-binding specificity differences alone could not explain genomic binding diversity of TFs from the ETS family. Authors proposed two possible mechanisms to achieve specificity for a certain family member: the divergent expression patterns of various family members and the cooperative binding of ETS factors with other TFs. The first mechanism was at least partially supported by (a) only partial overlapping of expression patterns of various family members revealed in transcriptome data [28] and (b) knock-down experiments replacing one member for another [29,30]. Results of our study and previous reviews on protein–protein interaction of ETS TFs with other TFs [31,32,33] strongly supported the second mechanism, i.e., combinatorial control of transcription as a characteristic property of ETS family members.

Outstanding properties of TFs from the ETS family were also supported by protein structure analysis [34,35,36,37,38,39,40,41,42,43]. In contrast to prokaryotes, the majority of eukaryotic TFs contained long stretches of intrinsically disordered regions (IDRs), which were sequences that did not adopt a stably structured conformation but they were essential for activity [44]. In TFs, IDRs were highly enriched around DNA binding domains (DBDs), which displayed electrostatically biased surfaces to their surroundings [45]. In the ETS family IDRs and highly stable α-helices flanking the DBD (ETS domain) were autoinhibitory for ETS1, ETS2, ETV6, ERG and ETV1/4/5 binding to DNA; ETS1, SPI1 and some other members of the ETS family were also regulated by another IDR serine-rich region [32,34,35,36,37]. DBD was autoinhibited in several family members by different mechanisms. Thus, a serine-rich IDR allosterically inhibited DNA binding of ETS1 through phosphorylation-enhanced interactions with the structured DBD and flanking N- and C-terminal inhibitory α-helices [38,39], or a single flanking C-terminal α-helix sterically inhibited DNA binding of ETV6 [34,40,41]. For ETV4 acetylation of selected lysines within the N-terminal IDR activated DNA binding, a C-terminal α-helix perturbed the conformation of its DNA-recognition helix [37]. Recently, experimental study of relatively distant paralogous ETS family members ETS1 and SPI1 has shown that the binding of DNA and the synthetic peptides containing IDRs by the DBD were mutually exclusive [42].

Thus, subfamily-specific α-helices that flank DBD and TF partners through IDRs could modify during TF–TF interaction the equilibrium between active and inactive states of a TF from ETS family; also, post-translational modifications within IDRs specifically regulated an individual ETS factor [37]. Hence, the regulatory strategy of TFs from ETS family consisted of activation through recruitment by other coactivators [43]. This conclusion is in good accordance with the results of our study.

Altogether, the results of our study allow one to improve the interpretation of ChIP-seq data and, accordingly, to clarify the understanding of functional interactions between TFs. We presume that the function of partner TFs does not consist of only indirect binding of anchor TFs (“tethering”); rather, the more conserved motifs of partner TFs may overlap less conserved motifs of anchor TFs. We propose the “permanent” model of cooperative binding of anchor and partner TFs (Figure 8), where various transition situations are possible. If an anchor TF binds genomic DNA directly, then the respective anchor motif was strongly conserved (Figure 8A). The presence of another TF (partner) may induce the protein–protein interaction anchor–partner that transforms this direct binding site of an anchor TF to CE anchor–partner with a more or less conserved anchor motif (Figure 8B,C), so that an anchor motif becomes moderately or weakly conserved, respectively. Finally, it is possible that an anchor TF loses even a weak contact with DNA, so that we may find in DNA only the motif of partner TF (Figure 8D, “tethering”).

Moreover, our findings can be helpful for the functional interpretation of GWAS noncoding SNPs and for revealing new regulatory variants. Recently, the prediction of potential TFBSs in ChIP-Seq data became the popular approach for the detection of genetic variants that were causal for various pathologies by affecting TF binding and gene regulation [46,47,48]. However, in this case numerous relatively weak (but causal) TF binding variants were usually missed and taking into account of cooperative TF binding via motifs co-occurrence was considered as one of the most promising approaches to resolve this issue [49].

## 4. Materials and Methods

### 4.1. MCOT: Classification of Co-Occurred Motifs

In the current study, we applied the MCOT package as described earlier [11] with some improvement (see below). This tool annotated pairs of overrepresented motifs, i.e., CEs. Input data of tool compiled peaks of a ChIP-seq dataset in the Fasta format, an anchor motif (nucleotide frequency matrix) with respect to potential BSs of the target TF, and either a partner motif or the list of partner motifs extracted from Hocomoco human or mouse core collections [14] (Figure 1A).

### 4.2. Composite Elements Search and Annotation

MCOT classified CEs according to the mutual orientation of motifs, e.g., for heterotypic CEs there were four distinct orientations (Figure 1B). There were three distinct cases of mutual locations: full/partial, overlaps and spacer, consequently MCOT used five computation flows (full, partial, overlap, spacer and any, Figure 1B). MCOT applied the recognition model of PWM for mapping motifs in peaks. For each matrix, five thresholds {T_1_, ..., T_5_} were used according to the unified set of expected FPRs for a whole genome dataset of promoters, {5.24 × 10^−5^, 1.02 × 10^−4^, 1.9 × 10^−4^, 3.33 × 10^−4^, 5 × 10^−4^}. The profile of the most stringent hits contained PWM scores T ≥ T_1_, the next profile comprised of scores in the range of T_2_ ≥ T > T_1_, etc. We estimated the conservation of a motif hit through an expected FPR as −Log_10_(FPR). For each of the 5 × 5 = 25 combinations of motifs conservations and each computation flow MCOT compiled the 2 × 2 contingency table (Table 1) and computed the significance of the Fisher’s exact test that compared the fractions of sequences with CEs and without them in peaks and background sequences, obtaining hits of both participating motifs. The background dataset was generated as described earlier [11].

MCOT subdivided all CEs into classes of asymmetric CEs with a more conservative anchor or partner motifs (Figure 1B). Hence, two additional Fisher’s tests estimated the enrichment of asymmetric CEs (Figure 1B), i.e., MCOT compared fractions of peaks/permuted sequences that contained only CEs with more conserved anchor or partner motifs, so that these calculations performed again according to Table 1.

The test of CE asymmetry (Table 2) implied for real and permuted sequences the comparison between counts of asymmetric CEs toward one and another motif.

### 4.3. Significances for Asymmetric CEs and for Asymmetry within CEs

We improved the MCOT algorithm [11] to calculate the asymmetry within CE as follows. We estimated the conservation of each motif by the expectation of its occurrence in the whole genome promoter dataset with the logarithmic measure −Log_10_(FPR). Than we applied the criteria

{−Log_10_[FPR(Anchor)] > −Log_10_[FPR(Partner)]} and,{−Log_10_[FPR(Anchor)] ≤ −Log_10_[FPR(Partner)]}.

Additionally, we classified all predicted CEs into two classes with a more conservative anchor or partner motifs. Next, for each class we computed the significance that compared counts of peaks containing/not containing CEs in the foreground and background datasets (Table 1). To estimate the asymmetry within CEs we applied the Fisher’s exact test that compared the count of CEs with more conserved anchor motifs and that for more conserved partner motifs in the foreground and background datasets (Table 2).

We assigned to the asymmetry significance −Log_10_[*p*-value] the sign “+” in the case of enrichment toward an anchor motif, otherwise, sign “−“ denoted the enrichment toward a partner motif. Next, for the foreground and background datasets of sequences we compiled the full lists of predicted CEs. We classified the conservation of each motif within the ranges of twelve conservation levels as follows [<3.5], [3.5; 3.7], [3.7; 3.9], etc., up to [5.3; 5.5] and [>5.5]. We computed the counts of CEs from foreground and background datasets Obs_i,j_ and Exp_i,j_ that had distinct combinations of conservation levels. Here indices *i* and *j* denote conservation levels for anchor and partner motifs. Finally, the per mille measure transforms the absolute CE counts to relative ones as follow: {1000 × Obs_i,j_/Obs)} and {1000 × Exp_i,j_/Exp}.

### 4.4. Bonferroni Correction for Significance

To take into account multiple comparisons we applied the Bonferroni’s correction and used the following critical values:
Significance of CEs regardless motifs conservation, 0.05/(N_FOR_ × N_BACK_ × N_FLOW_ × N_THR_ × N_THR_);Significance of asymmetric CEs toward one of motifs, 0.05/(N_FOR_ × N_BACK_ × N_FLOW_ × 2);CE asymmetry, 0.05/(N_FOR_ × N_BACK_ × N_FLOW_).


Here N_FOR_ and N_BACK_ means the size of the foreground and background datasets (i.e., the number of peaks and random sequences, which generated in MCOT [11], N_FLOW_ = 5 designates the number of MCOT computation flows and N_THR_ = 5 means the number of thresholds for each motif).

### 4.5. Massive Analysis of the ChIP-Seq Data

In the current study, we complemented previously published benchmark ChIP-seq data [11] for human TFs, so the whole collection consisted of 119 ChIP-seq datasets for 45 TFs (Appendix A). As in an earlier study [11], for each dataset we annotated the results of the de novo motif search [8], manually selected enriched motifs with respect to the anchor TF and approved the homology between the de novo detected and known motifs [50]. We applied the MCOT as described earlier and above in this study [11,51]. In particular, 396 partner motifs of human TFs were extracted from the Hocomoco human core collection [14,52] (Figure 1A). We used the classification of human and mouse TFs according to the characteristics of their DNA-binding domains [13,53] (Figure 1A). We supplied all partner motifs with the names of respective families and classified all motifs into 67 distinct families of TFs. Since the consequent analysis was based on the recognition of motifs, we performed the pairwise comparison of homology of all partner motifs with the motif comparison tool from the MCOT [11] (*p* < 0.05 for at least one of two motifs similarity measures). In our analysis we preserved the classification of motifs according to their families [13], but in specific cases we annotated together homologous motifs from various families. In particular, according to previous data [12] we distinguished the following groups of motifs:
Jun-like, out of a total 18 motifs of Jun-related {1.1.1}, Fos-related {1.1.2} and Maf-related {1.1.3} families 15 were homologous;ETS-like, out of a total 19 motifs of the ETS-related factors {3.5.2} family 14 were homologous;CTCF-like, two homologous motifs constituted the subfamily CTCF-like factors {2.3.3.50} of the largest family More than three adjacent zinc finger factors {2.3.3} consisting of 76 motifs;Two non-homologous motifs THA11_HUMAN.H11MO.0.B and THAP1_HUMAN.H11MO.0.C constituted the THAP-related factors {2.9.1} family.


We also considered the following motifs, classified according to the TF families:
p53-like, all three motifs from family p53-related factors {6.3.1} were homologous;RFX-like, all four motifs from family RFX-related factors {3.3.3} were homologous;GATA-like, all five motifs from family GATA-type zinc fingers {2.2.1} were homologous, we added them to their homologue TAL1_HUMAN.H11MO.0.A from the Tal-related factors {1.2.3} family (the rest of the participants of this family were not homologous to GATA-like motifs);NR1H3-like motifs, four motifs from the thyroid hormone receptor-related factors (NR1) {2.1.2} family (NR1H3_HUMAN.H11MO.0.B, THA_HUMAN.H11MO.0.C, NR1I3_HUMAN.H11MO.0.C and NR1I2_HUMAN.H11MO.0.C) were homologous, this family consisted of 14 motifs; NR1H3-like motifs had close homologous motifs in families of steroid hormone receptors (NR3) {2.1.1} (e.g., ERR1_HUMAN.H11MO.0.A) and RXR-related receptors (NR2) {2.1.3} (e.g., COT2_HUMAN.H11MO.0.A);NFYA-like, all three motifs from the family heteromeric CCAAT-binding factors {4.2.1} were homologous.


We selected for consequent analysis 49 families with at least two motifs among all 67 families respecting all 396 partner motifs. We also included in analysis the CTCF-like factors {2.3.3.50} subfamily, since CTCF-like motifs were previously annotated [12]. Thus, we included in the analysis 50 clades of partner TFs, including 49 families and one subfamily.

We performed the prediction of potential CEs with the MCOT for the benchmark data of 119 ChIP-seq datasets (Appendix A). We proposed that homology between an anchor and partner motifs might influence CEs enrichment. Hence, we excluded CEs consisting of significantly similar partner and anchor motifs. We presumed the significant similarity if at least one of two motifs similarity measures used showed the significant similarity (*p* < 0.05, [11]). We applied Bonferroni’s correction for the significance of CEs (see above) and counted ChIP-seq datasets with significant CEs separately for five MCOT computation flows.

We used the MEGA package to draw trees that showed the similarity of motifs [54,55].

## 5. Conclusions

We proposed the approach for the computation of the significance of co-occurrence of asymmetric CEs anchor–partner with one of the participant motifs more conservative than another one, and for asymmetry within pairs of co-occurred motifs;We applied our approach for motifs of partner TFs from various families over-represented near motifs of anchor TFs in ChIP-seq data;We demonstrated that for partner motifs of almost all families of TFs only for overlapping anchor–partner pairs but not for pairs with a spacer, pairs with a higher conservation of partner motifs were significantly more abundant than those with higher conservation of anchor motifs. This observation explained a substantial portion of ChIP-seq data lacking conserved anchor motifs;We found that the asymmetric CEs toward partner motifs were the most reliable for partner motifs of TFs from ETS family. Hence, motifs of TFs from the ETS family tended to mediate the interaction of anchor TFs with genomic DNA.

## Figures and Tables

**Figure 1 ijms-21-06023-f001:**
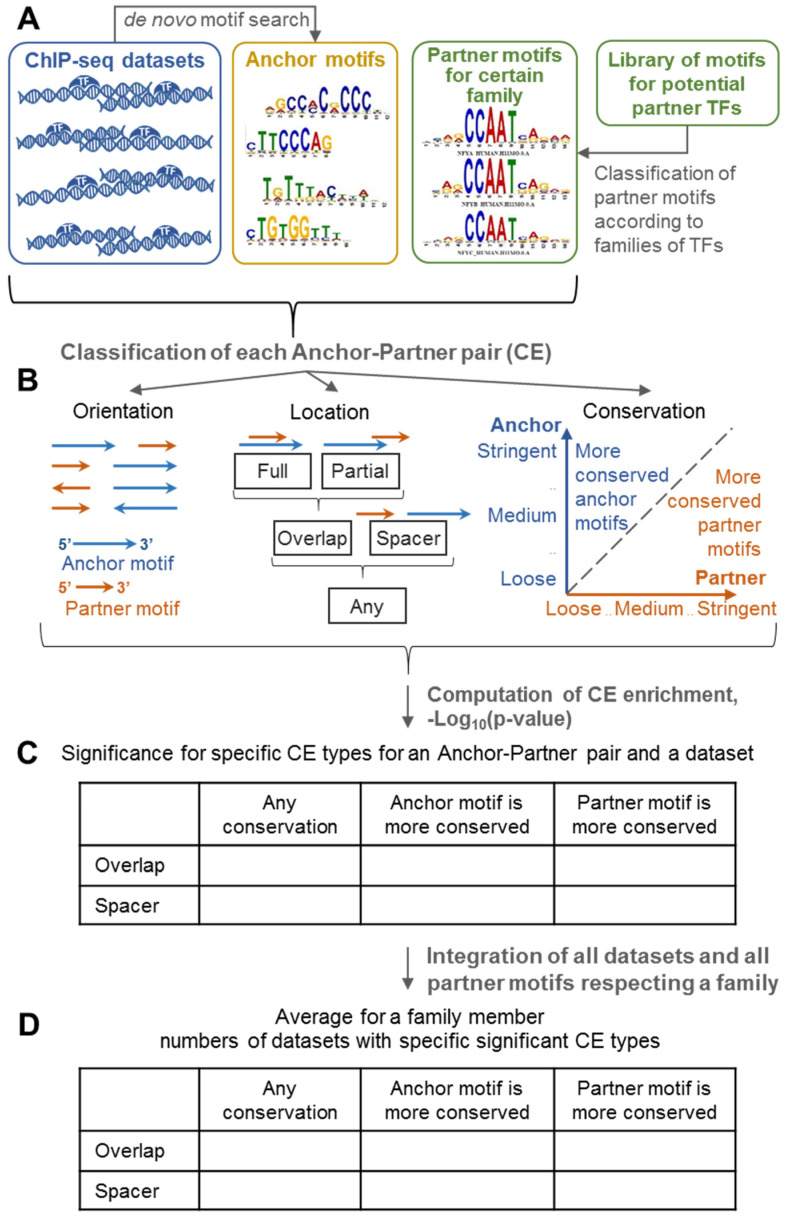
The workflow of the motifs co-occurrence tool (MCOT) application in the current study. Basic input data preparation comprises of an application of the de novo motif search tool [8] for a collection of ChIP-seq datasets, and classification of partner motifs from the public library [14] into families according to the structure of the DNA-binding domain [13] (**A**). Next, MCOT performs composite elements (CEs) classification according to orientations, overlaps or spacers and relationships of motifs conservation (**B**); MCOT computes significances of enrichment for various CE types, so that Bonferroni’s correction is applied (**C**) (Section 4.4). Finally, average counts of ChIP-seq datasets possessing certain CE type for all family members reflect their common tendency to participate in specific CEs in the benchmark ChIP-seq data (**D**).

**Figure 2 ijms-21-06023-f002:**
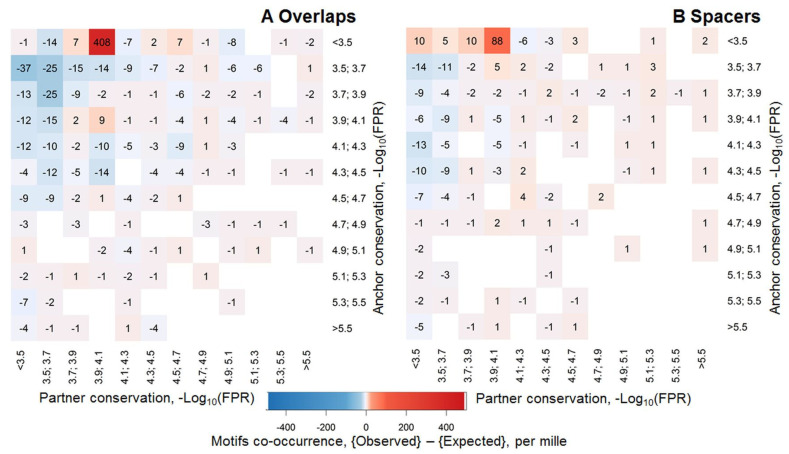
The difference between observed and expected abundances of CEs with specific conservation of the anchor FoxA2 (axis Y) and partner HNF1β (axis X) motifs for ChIP-seq data [15] in per mille. The conservation of motifs was measured as −Log_10_(FPR) (logarithmic false positive rate (FPR), Section 4.2). The color on both heatmaps shows the difference between observed (peaks) and expected (permuted sequences) relative abundance of CEs with specific conservation levels (Section 4.3). FoxA2 and HNF1β motifs were derived from the Homer de novo motif search [8] and Hocomoco database (HNF1B_Mouse.H11MO.0.A) [14], respectively. Panels (**A**) and (**B**) show asymmetry of CEs with an overlap of motifs and with a spacer, respectively.

**Figure 3 ijms-21-06023-f003:**
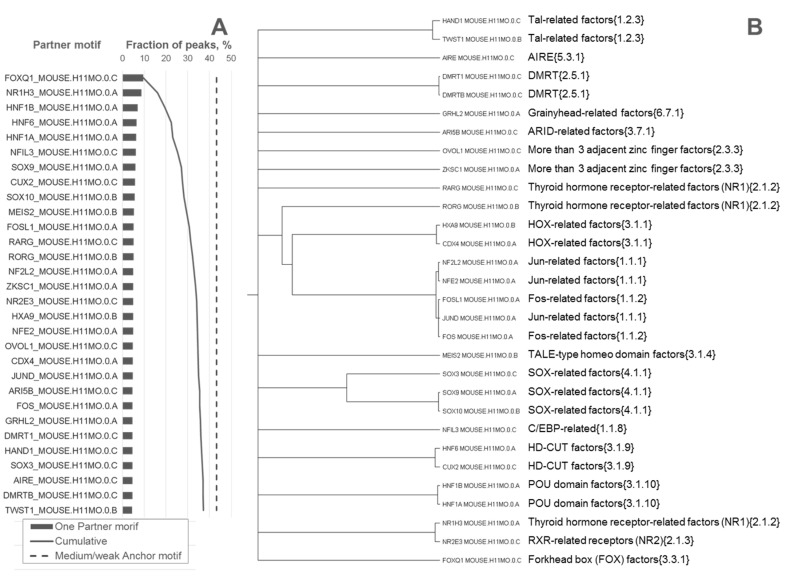
The analysis of FoxA2 peaks [15] that contained the FoxA2 motifs of moderate and weak conservation. Panel (**A**) displays fractions of analyzed peaks that contained asymmetric CEs with specific partner motifs. Panel (**B**) shows the tree of similarity for the selected list of the 30 top-ranked partner motifs from panel A and the respective families of partner transcription factors (TFs) [13]. We took in an analysis of only 43.21% of all peaks with the best scores of peaks in the range of FPR from 5.24 × 10^−5^ to 5 × 10^−4^, see the dashed line in panel (**A**). We applied MCOT package and defined the top-ranked 30 partner motifs from the Hocomoco mouse core collection [14] that did not have a similarity to the anchor motif (*p* < 0.05, any similarity measure from [11]) and respected the significant asymmetric CEs toward the partner motif. We sorted partner motifs according to the fraction of peaks that contained such asymmetric CEs and computed the cumulative fraction of peaks that contained at least one, two, three, etc., up to 30 types of CE types with various top-ranked partner motifs, see the regular line in panel (**A**).

**Figure 4 ijms-21-06023-f004:**
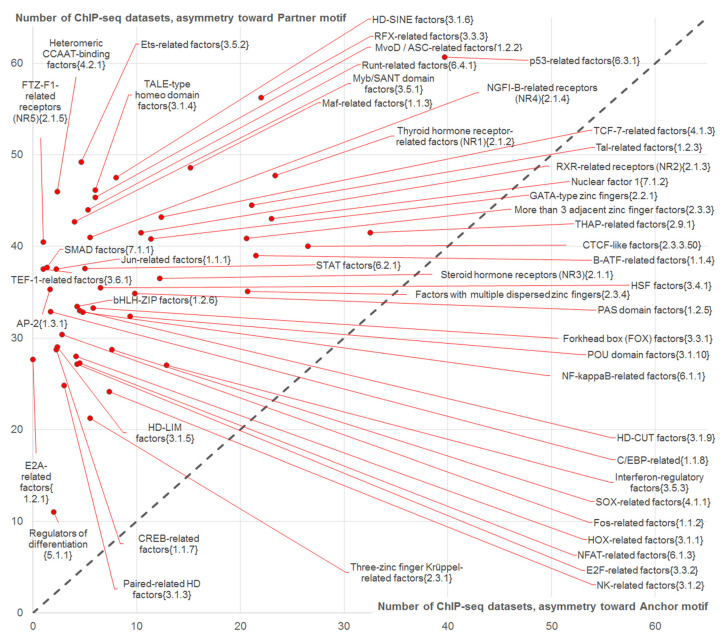
The scatterplot of abundances of asymmetric CEs toward the anchor (axis X) and asymmetric CEs toward the partner (axis Y) motifs for 50 clades of TFs. These clades comprised of 49 families of TFs with at least two motifs and subfamily CTCF-like factors {2.3.3.50} with two motifs from the Hocomoco human core collection [14]. Total number of ChIP-seq datasets was equal to 119 (Appendix A). Only CEs with an overlap of motifs were considered. The diagonal dashed line marks equal numbers of datasets, it implies the partitioning of all clades into those with the higher abundance of asymmetric CEs toward partner motifs (top left triangle, all 50 clades) and those with the higher abundance of asymmetric CEs toward anchor motifs (bottom right triangle without clades).

**Figure 5 ijms-21-06023-f005:**
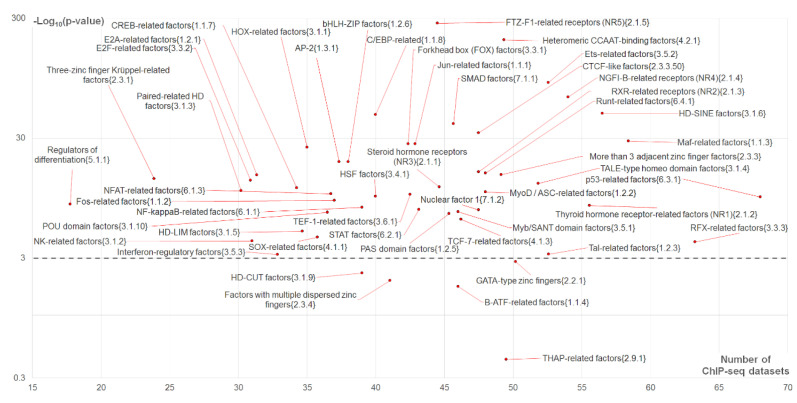
The significance of enrichment of asymmetric CEs toward the partner motifs as a function of CE abundance. The scatterplot shows 50 clades of partner TFs, including 49 families of TFs with at least two motifs and subfamily CTCF-like factors {2.3.3.50} with two motifs from the human core Hocomoco collection [14]. The total number of ChIP-seq datasets is 119 (Appendix A). Axis X implies the number of ChIP-seq datasets with predicted CEs with an overlap of anchor motifs and partner motifs from a specific clade and without taking into account motifs conservation. Axis Y shows the significance of the Welch’s *t*-test that for each clade compare the number of datasets containing asymmetric CEs toward partner motifs and overlaps of motifs and the respective number of datasets containing asymmetric CEs toward anchor motifs. The horizontal dashed line marks Bonferroni’s correction for the *t*-test significance, −Log_10_(*p*-value) = 3.

**Figure 6 ijms-21-06023-f006:**
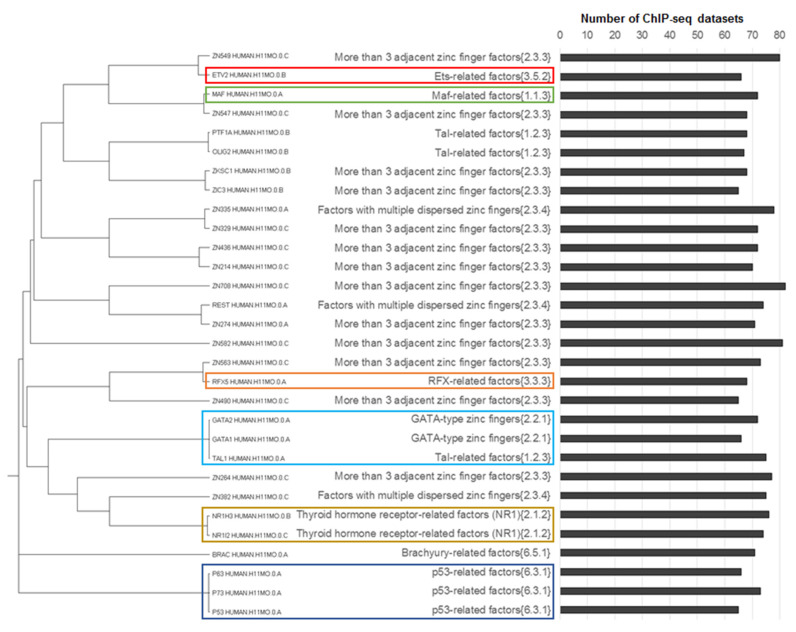
Clustering of 30 top-ranked partner motifs from the Hocomoco human core collection [14], according to their abundance in CEs predicted with an overlap of anchor motifs. We excluded from the analysis CEs containing the significant homology between motifs. The left/middle/right columns show the tree constructed according to motifs homology, names of TF families [13] and the distribution of the number of ChIP-seq datasets that contained respective CEs. Brown, green, red, orange, blue and aqua boxes mark NR1H3-like motifs from thyroid hormone receptor-related factors (NR1) {2.1.2} family, Jun-like (Maf-related factors {1.1.3}), ETS-like (ETS-related factors {3.5.2}), RFX-like (RFX-related factors {3.3.3}, p53-like (p53-related factors {6.3.1}) and GATA-like (Tal-related factors {1.2.3}) motifs, respectively. Totally, we included in the analysis 119 ChIP-seq datasets for human TFs (Appendix A).

**Figure 7 ijms-21-06023-f007:**
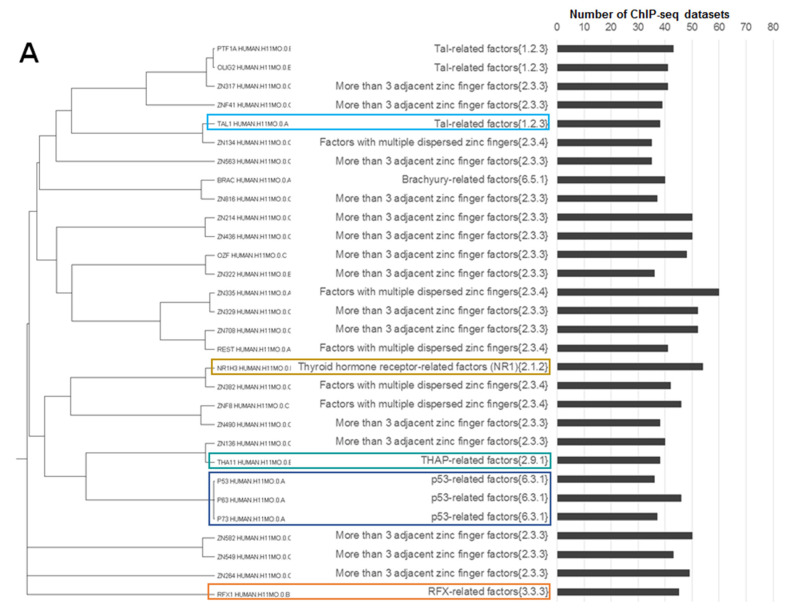
Clustering of the 30 top-ranked partner motifs from the Hocomoco human core collection [14] according to their abundance in CEs predicted with an overlap of anchor motifs. We excluded from the analysis CEs containing the significant homology between motifs. Panels (**A**,**B**) show results for CEs with more conserved anchor and partner motifs, respectively. For each panel the left/middle/right columns show the tree constructed according to motifs homology, names of TF families [13] and the distribution of the number of ChIP-seq datasets that contained the respective CEs. Brown, green, red, orange, blue, cyan and aqua boxes mark NR1H3-like motifs from the thyroid hormone receptor-related factors (NR1) {2.1.2} family, Jun-like (Maf-related factors {1.1.3}), ETS-like (ETS-related factors {3.5.2}), RFX-like (RFX-related factors {3.3.3}, p53-like (p53-related factors {6.3.1}), THAP-related factors {2.9.1} and GATA-like (Tal-related factors {1.2.3}) motifs, respectively. Totally, we included in the analysis 119 ChIP-seq datasets for human TFs (Appendix A).

**Figure 8 ijms-21-06023-f008:**
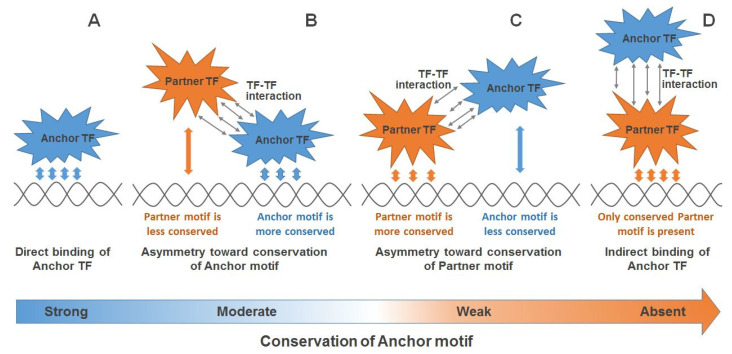
The “permanent” model of cooperative binding of an anchor and partner TFs for the explanation of a substantial portion of ChIP-seq data lacking conserved motifs of anchor TFs. Panel (**A**) is in respect to the most conserved motifs of an anchor TF in a ChIP-seq dataset, such motifs are in most cases overrepresented and successively recognized as the canonical motif of anchor TF. However, an anchor TF often participates in TF–TF interactions with multiple partner TFs. Thus, a whole conservation of anchor–partner CE is subdivided between anchor and partner motifs. We propose here two options: an anchor motif preserves the higher conservation than a partner motif (**B**), or an anchor motif has less conserved motif than a partner motif (**C**). Finally, an anchor TF binds to DNA indirectly (**D**), e.g., if a heterodimer of the anchor/partner TFs binds with DNA only through partner TF. The long arrow in the bottom reflects the permanent decrease/increase of the conservation of an anchor/partner motif. Numbers of red/blue arrows between each TF and DNA reflect the conservation of the respective motif.

**Table 1 ijms-21-06023-t001:** 2 × 2 contingency tables for the calculation of the significance of CE. We applied this table for computation of the CE significance regardless motifs conservation (this is in respect to all CEs of the scatterplot in Figure 1B) and CE significances for asymmetric CE with more conserved one or another motif (these cases are in respect to two triangles of the same scatterplot).

Categories of Sequences	Count of Sequences	
	With CE	Without CE
Foreground	Obs_CE+_	Obs_CE-_
Background	Exp_CE+_	Exp_CE-_

**Table 2 ijms-21-06023-t002:** 2 × 2 contingency tables for calculation of CE asymmetry.

Categories of CEs	Count of CEs with More Conserved	
	Anchor Motif	Partner Motif
Foreground	Obs_CE,Anchor_	Obs_CE,Partner_
Background	Exp_CE,Anchor_	Exp_CE,Partner_

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
