# Peer review of "Asymmetric Conservation within Pairs of Co-Occurred Motifs Mediates Weak Direct Binding of Transcription Factors in ChIP-Seq Data"

_ijms, 2020, doi:10.3390/ijms21176023_

Round 1

Reviewer 1 Report

The article of Victor Levitsky and colleagues entitled “Asymmetric conservation within pairs of co-occurred motifs mediates weak direct transcription factor binding in CHIP-seq data” is a very interesting work. It is a pleasure to read this well written manuscript. Therefore, the manuscript can be published in the current version after minor revision in the International Journal of Molecular Sciences.

My minor concerns are:

- The authors should delete “followed by” in line 25.

- The authors have to prove all their statements with citations in the Introduction and Discussion part. Citations are missing in lines 32, 35, 49, 50,497, 498, 500 and 513.

- The authors have to be more precise in line 490; the number of ETS family members is well known.

Author Response

Point 1

  • The authors should delete “followed by” in line 25.

Reply

Corrected. We carefully checked all manuscript and corrected it

Point 2

  • The authors have to prove all their statements with citations in the Introduction and Discussion part. Citations are missing in lines 32, 35, 49, 50,497, 498, 500 and 513.

Reply

All statements were proved with citations

Point 3

  • The authors have to be more precise in line 490; the number of ETS family members is well known.

Reply

We inserted the respective number according to the recent review

Reviewer 2 Report

Levitsky and colleagues presented a computational study aimed at elucidating the interconnection between transcription factors and their binding sites with particular references to the overlapped motifs where multiple TFs form composite elements. For their purpose, the authors used a complex algorithm mainly based on a previous computational tool, the Motifs Co-Occurrence Tool (MCOT) used for the analysis of ChIP-seq dataset and able to predict CEs. Overall, the manuscript is well written, however, the description of the sections is too verbose and difficult to understand for scholars interested in the study of transcription factors. Below are reported some comments that will improve the quality of the manuscript:

1) The authors mechanistically describe their manuscript and results. However, due to the wide field of application of transcription factors, it could be useful to emphasize the translational impact of the approach here proposed highlighting the potential application of the analysis of CEs in different pathologies. Add information on studies already performing such analysis could be useful;

2) The manuscript is too verbose and there are many alliterations (e.g. line 62-64; 70-72; etc.). Please consider to concisely describe the different parts of the manuscript;

3) The authors should provide a schematic representation of the algorithm or of the workflow used in order to facilitate the understanding of the study.

Author Response

Point 1

The authors mechanistically describe their manuscript and results. However, due to the wide field of application of transcription factors, it could be useful to emphasize the translational impact of the approach here proposed highlighting the potential application of the analysis of CEs in different pathologies. Add information on studies already performing such analysis could be useful;

Reply

We carefully corrected the text through all sections. I particular, at the end of Discussion section  we added the new piece of text with several references, this piece concluded the discussion, proposed possible application of our research in different pathologies

Point 2

The manuscript is too verbose and there are many alliterations (e.g. line 62-64; 70-72; etc.). Please consider to concisely describe the different parts of the manuscript;

Reply

We carefully corrected the text through all sections. We tried to reach more clear and understandable style.

Point 3

The authors should provide a schematic representation of the algorithm or of the workflow used in order to facilitate the understanding of the study.

Reply

We almost completely changed the content of Figure 1. Now it reflects the workflow of our study. All respective parts of the manuscript were also rewritten